# Query-Based Adversarial Prompt Generation

**Jonathan Hayase**[1]    **Ema Borevkovic**[2]    **Nicholas Carlini**[3]    **Florian Tramèr**[2]    **Milad Nasr**[3]

[1]University of Washington    [2]ETH Zürich    [3]Google Deepmind

## Abstract

Recent work has shown it is possible to construct adversarial examples that cause aligned language models to emit harmful strings or perform harmful behavior. Existing attacks work either in the white-box setting (with full access to the model weights), or through *transferability*: the phenomenon that adversarial examples crafted on one model often remain effective on other models. We improve on prior work with a *query-based* attack that leverages API access to a remote language model to construct adversarial examples that cause the model to emit harmful strings with (much) higher probability than with transfer-only attacks. We validate our attack on GPT-3.5 and OpenAI's safety classifier; we can cause GPT-3.5 to emit harmful strings that current transfer attacks fail at, and we can evade the OpenAI and Llama Guard safety classifiers with nearly 100% probability.

## 1   Introduction

The rapid progress of transformers [33] in the field of language modeling has prompted significant interest in developing strong *adversarial examples* [4, 30] that cause a language model to misbehave harmfully. Recent work [36] has shown that by appropriately tuning optimization attacks from the literature [28, 14], it is possible to construct adversarial text sequences that cause a model to respond in a targeted manner.

These attacks allow an adversary to cause an otherwise "aligned" model—that typically refuses requests such as "how do I build a bomb?" or "swear at me!"—to comply with such requests, or even to emit *exact* targeted unsafe strings (e.g., a malicious plugin invocation [3]). These attacks can cause various forms of harm, ranging from reputational damage to the service provider, to potentially more significant harm if the model has the ability to take actions on behalf of users [13] (e.g., making payments, or reading and sending emails).

The class of attacks introduced by Zou et al. [36] are white-box optimization attacks: they require complete access to the underlying model to be effective—something that is not true in practice for the largest production language models today. Fortunately (for the adversary), the *transferability* property of adversarial examples [26] allows an attacker to construct an adversarial sequence on a local model and simply replay it on a larger production model to great effect. This allowed Zou et al. [36] to fool GPT-4 and Bard with 46% and 66% attack success rate by transferring adversarial examples initially crafted on the Vicuña [11] family of open-source models.

**Contributions.**   In this paper, we design an optimization attack that *directly* constructs adversarial examples on a remote language model, without relying on transferability.[1] This has two key benefits:

- Targeted attacks: Query-based attacks can elicit specific harmful outputs, which is not feasible for transfer attacks.

---

[1]There exist other black-box jailbreak methods that rely on either a language model to refine candidate attacks [22, 8] or on greedy search [2]. These techniques are weaker than ours though, and do not succeed in making a target model output exact harmful strings, which we do.

38th Conference on Neural Information Processing Systems (NeurIPS 2024).

- Surrogate-free attack: Query-based attacks also allow us to generate adversarial text sequences when no convenient transfer source exists.

Our fundamental observation is that each iteration of the GCG attack of Zou et al. [36] can be split into two stages: filtering a large set of potential candidates with a gradient-based filter, followed by selecting the best candidate from the shortlist using only query access. Therefore, by replacing the first stage filter with a filter based on a surrogate model, and then directly querying the remote model we wish to attack, we obtain a query-based attack which may be significantly more effective than an attack based only on transferability.

We further show how an optimization to the GCG attack allows us to remove the dependency on the surrogate model completely, with only a moderate increase in the number of model queries. As a result, we obtain an effective query-only attack requiring no surrogate model at all.

As an example use-case, we show how to evade OpenAI's content moderation endpoint (that, e.g., detects hateful or explicit sentences) with nearly 100% attack success rate without having a local content moderation model available. This is despite this endpoint being OpenAI's "most robust moderation model to-date" [24].

## 2 Background

**Adversarial examples.** First studied in the vision domain, adversarial examples [4, 30] are inputs designed by an adversary to make a machine learning model misbehave. Early work focused on the "white-box" threat model, where an adversary has access to the model's weights and can thus use gradient descent to reliably maximize the model's loss with minimum perturbation [12, 6, 21].

These attacks were then extended to the more realistic "black-box" threat model, where an adversary has no direct access to the model weights. The first black-box attacks relied on the "transferability" of adversarial examples [26]: attacks that fool one model also tend to fool other models trained independently—even on different datasets.

Transfer-based attacks have several limitations. Most importantly, they rarely succeed at "targeted" attacks that aim to cause a model to perform a *specific* incorrect behavior. Even on simple tasks like ImageNet classification with 1,000 classes, targeted transfer attacks are challenging [20].

These difficulties gave rise to *query-based* black-box attacks [9, 5]. Instead of relying exclusively on transferability, these attacks query the target model to construct adversarial examples using black-box optimization techniques. These attacks have (much) higher success rates: they can reach nearly 100% targeted attack success rate on black-box ImageNet classifiers, at a cost of a few thousand model queries. Query-based attacks can further be combined with signals from a local model to reduce the number of model queries without sacrificing attack success [10].

**Language models.** Language models are statistical models that learn the underlying patterns within text data. They are trained on massive datasets of text to predict the probability of the next word or sequence of words, given the preceding context. These models enable a variety of natural language processing tasks such as text generation, translation, and question answering [27].

**NLP adversarial examples.** Adversarial examples for language models have followed a similar path as in the vision field. However, due to the discrete nature of text, direct gradient-based optimization is more difficult. Early work used simple techniques such as character-level or word-level substitutions to cause models to misclassify text [18].

Further attacks optimized for adversarial text in a language model's continuous *embedding space*, and then used heuristics to convert adversarial embeddings into hard text inputs [28]. These methods, while effective on simple and small models, were not sufficiently strong to reliably cause errors on large transformer models [7]. As a result, followup work was able to combine multiple ideas from the literature in order to improve the attack success rate considerably [36].

In doing so, Zou et al. [36] also introduced the first set of transferable adversarial examples that were also capable of fooling multiple production models. By generating adversarial examples on Vicuna—a freely accessible large language model with open weights—it was possible to construct transferable adversarial examples that fool today's largest models, including GPT-4.

Unfortunately, NLP transfer attacks suffer from the same limitations as their counterparts in vision:

- Transfer attacks require a high-quality surrogate. For example, Zou et al. [36] showed that Vicuña is a poor surrogate for Claude, achieving just 2% transfer attack success rate.

- Transfer attacks do not succeed at inducing targeted "harmful strings". While transfer attacks can cause models to comply with requests (i.e., an un-targeted attack), they cannot force the model into producing a specific harmful output.

As we will show, it is possible to address both of these limitations (and more!) through query-based attacks (in concurrent work, Sitawarin et al. [29] propose similar attacks to ours, but do not evaluate them for inducing targeted harmful strings).

**The Greedy Coordinate Gradient attack (GCG).**  Zou et al. [36] recently proposed an extension of the AutoPrompt method [28], known as Greedy Coordinate Gradient (GCG), which has proven to be effective. GCG calculates gradients for all possible single-token substitutions and selects promising candidates for replacement. These replacements are then evaluated, and the one with the lowest loss is chosen. Despite its similarity to AutoPrompt, GCG significantly outperforms it by considering all coordinates for adjustment instead of selecting only one in advance. This comprehensive approach allows GCG to achieve better results with the same computational budget. Zou et al. also optimized the adversarial tokens for several prompts and models at the same time, which helps to improve the transferability of the adversarial prompt to closed source models.

**Heuristic approaches.**  Given the popularity of language models, many also craft adversarial prompts by manually prompting the language models until they produced the desired (harmful) outputs. Inspired by manual adversarial prompts, recent works showed that they can improve the manual style attacks using several heuristics [35, 15, 34].

## 3 GCQ: Greedy Coordinate Query

We now introduce our attack: Greedy Coordinate Query (GCQ). At a high level, our attack is a direct modification of the GCG method discussed above.

### 3.1 Method

Our main attack strategy is similar to GCG in that it makes greedy updates to an adversarial string. At each iteration of the algorithm, we perform an update based on the best adversarial string found so far, and after a fixed number of iterations, return the best adversarial example.

The key difference in our algorithm is in how we choose the updates to apply. Whereas GCG maintains exactly one adversarial suffix and performs a brute-force search over many potential updates, to increase the query efficiency, our update algorithm is reminiscent of best-first-search. Each "node" corresponds to a given adversarial suffix. Our attack maintains a buffer of the $B$ best unexplored nodes. At each iteration, we take the best node from the buffer and expand it. The expansion is done by sampling a large set of $b_p$ neighbors, taking the $b_q$ best of those according to a local proxy loss, $\ell_p$ and evaluating

---

**Algorithm 1:** Greedy Coordinate Query

**input** : vocabulary $V = \{v_i\}_{i=1}^n$, sequence length $m$, loss $\ell : V^m \to \mathbb{R}$, proxy loss $\ell_p : V^m \to \mathbb{R}$, iteration count $T$, proxy batch size $b_p$, query batch size $b_q$, buffer size $B$

$\texttt{buffer} \leftarrow B$ uniform samples from $V^m$
**for** $i \in [T]$ **do**
    **for** $i \in [b_q]$ **do**
        $j \sim \text{Unif}([m]), t \sim \text{Unif}(V)$
        $\texttt{batch}_i \leftarrow \text{argmin}_{b \in \texttt{buffer}}\, \ell(b)$
        $(\texttt{batch}_i)_j \leftarrow t$
        $\texttt{ploss}_i \leftarrow \ell_p(\texttt{batch}_i)$
    **end**
    **for** $i \in \text{Top-}b_q(\texttt{ploss})$ **do**
        $\texttt{loss} \leftarrow \ell(\texttt{batch}_i)$
        $b_{\text{worst}} \leftarrow \text{argmax}_{b \in \texttt{buffer}}\, \ell(b)$
        **if** $\texttt{loss} \leq \ell(b_{\text{worst}})$ **then**
            remove $b_{\text{worst}}$ from $\texttt{buffer}$
            add $\texttt{batch}_i$ to $\texttt{buffer}$
        **end**
    **end**
**end**
**return** $\text{argmin}_{b \in \texttt{buffer}}\, \ell(b)$

---

these with the true loss $\ell$. We then iterate over the neighbors and update $B$. We write the algorithm in pseudocode in Algorithm 1.

In practice, $\texttt{buffer}$ is implemented using a min-max heap containing pairs of examples and their corresponding losses (with order defined purely by the losses). This allows efficient read-write access to both the best and worst elements of the buffer.

Following [36], we use the negative cumulative logprob of the target string conditioned on the prompt as our loss $\ell$. For our proxy loss, we use the same loss but evaluated with a local proxy model instead. We consider the attack to be successful if the target string is generated given the prompt under greedy sampling.

## 3.2 Practical considerations

### 3.2.1 Scoring prompts with logit-bias and top-5 logprobs

Around September 2023, OpenAI removed the ability to calculate logprobs for tokens supplied as part of the prompt. Without this, there was no direct way to determine the cumulative logprob of a target string conditioned on a prompt. Fortunately, the existing features of the API could be combined to reconstruct this value, albeit at a higher cost. We describe the approach we used to reconstruct the logprobs, which is similar to the technique proposed in [23], in Appendix B. This is the method we used for our OpenAI harmful string results. Later, in March 2024, OpenAI further updated their API so that the `logit_bias` parameter does not affect the tokens returned by `top_logprobs`. As of May 2024, it is still possible to infer logprobs using the binary search procedure of [23], although the resulting attack will be significantly more expensive.

### 3.2.2 Short-circuiting the loss

The method described previously calculates the cumulative logprob of a target sequence conditioned on a prompt by iteratively computing each token's contribution to the total. In practice, we can exit the computation of the cumulative logprob early if we know it is already sufficiently small. This was the main motivation for the introduction of the buffer. Because we maintain a buffer of the $B$ best unexplored prompts seen so far, we know that any prompt with a loss greater than $\ell(b_{\text{worst}})$ will be discarded. In practice, we find this optimization reduces the total cost of the attack by approximately 30%.

### 3.2.3 Choosing a better initial prompt

In Algorithm 1, we initialize `buffer` with uniform random $m$-token prompts. However, in practice, we found it is better to initialize the buffer with a prompt that is designed specifically to elicit the target string. In particular, we found that simply repeating the target string as many times as the sequence length allows, truncating on the left, to be an effective choice for the initial prompt. This prompt immediately produces the target string (without needing to run Algorithm 1) for 28% of the strings in harmful strings when $m = 20$. We perform an ablation study of this initialization technique in Section 4.3.

## 3.3 Proxy-free query-based attacks

The attacks we described so far rely on a local proxy model to guide the adversarial search. As will see, such proxies may be available even if there is no good surrogate model for transfer-only attacks. Yet, there are also settings where an attacker will not have access to good proxy models. In this section, we explore the possibility of *pure query-based attacks* on language models.

We start from the observation that in existing optimization attacks such as GCG, the model gradient provides a rather weak signal (this is why GCG combines gradients with greedy search). We can thus build a simple query-only attack by ignoring the gradient entirely; this leads to a purely greedy attack that samples random token replacements and queries the target's loss to check if progress has been made. However, since the white-box GCG attack is already quite costly, the additional overhead from foregoing the gradient information can be prohibitive.

Therefore, we introduce a further optimization to GCG, which empirically reduces the number of model queries by a factor of $2\times$. This optimization may be of independent interest. Our attack variant differs from GCG as follows: in the original GCG, each attack iteration computes the loss for $B$ candidates, each obtained by replacing the token in one random position of the suffix. Thus, for a suffix of length $l$, GCG tries an average of $B/l$ tokens in each position. We instead focus our search on a single position of the adversarial suffix. Crucially, instead of choosing this position at random as in AutoPrompt, we first try a single token replacement in each position, and then write down the position where this replacement reduced the loss the most. We then try $B'$ additional token

replacements for just that one position. In practice, we can set $B' \ll B$ without affecting the attack success rate.

## 4 Evaluation

We now evaluate four aspects of our attack:

1. In Section 4.1 we evaluate the success rate of a modified GCG on open-source models, allowing us to compare to the white-box attack success rates as a baseline.
2. In Section 4.3 we evaluate how well GCQ is able to cause production language models like `gpt-3.5-turbo` to emit harmful strings, something that transfer attacks alone cannot achieve.
3. In Section 4.4, we evaluate the effectiveness of the proxy-free attack described in Section 3.3.
4. Finally, in Section 4.5 we develop attacks that fool the OpenAI content moderation model; these attacks test our ability to exploit models without a transfer prior.

### 4.1 Harmful strings for open models

We give transfer results for aligned open source models using GCG. Unlike the transfer results in [36], we maintain query access to the target model, but replace the model gradients with the gradients of a proxy model. We tuned the parameters to maximize the attack success rate within our compute budget, since we are not limited by OpenAI pricing. We used a batch size of 512 and a maximum number of iterations of 500. This corresponds to nearly 400 times more queries than we allow for the closed models in Section 4.3.

First, to establish a baseline, we report results for white-box attacks on Vicuna [11] version 1.3 which is fine-tuned from Llama 1 [31] as well as Llama 2 Chat [32] in Figure 1a. Here we see that the Vicuna 1.3 models become more difficult to attack as their scale increases, and the smallest Llama 2 model is significantly more resistant than even the largest Vicuna model.

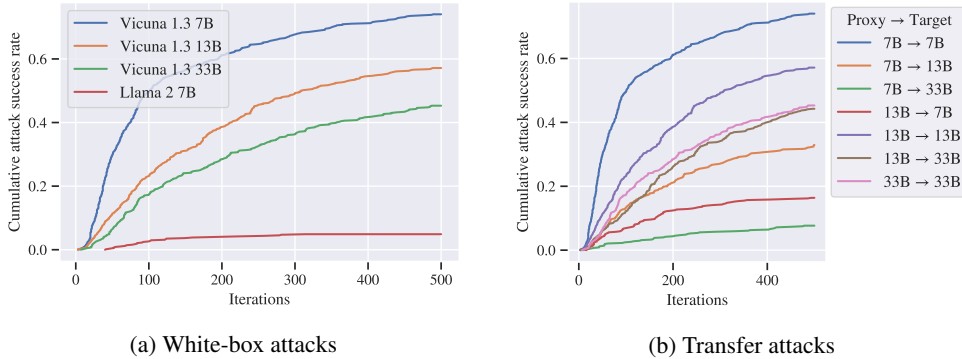

(a) White-box attacks                      (b) Transfer attacks

Figure 1: Harmful strings for open models. We show white-box results in (a), where we see Llama-2 is more robust than Vicuna. In (b), we show transfer attacks within the Vicuna 1.3 model family, where we see that transfer attacks are most successful when the models are of similar size.

We give results for transfer between scales within the Vicuna 1.3 model family in Figure 1b. Interestingly, we find that the 7B model transfers poorly to larger scales, while there is little loss transferring 13B to 33B. On the other hand, 13B transfers poorly to 7B. This suggests that the 13B and 33B models are more similar to each other than they are to 7B.

### 4.2 Comparison to other attacks

For the sake of comparison, we modify AutoDAN [19] to perform the harmful strings attack in the same setting as our experiments in Section 4.1. We include these results to demonstrate that harmful string are more difficult to elicit than jailbreaks, and that even highly effectively jailbreaking attacks are not automatically able to perform harmful string attacks.

In AutoDAN's original setting, a jailbreaking attack is considered successful if the model generates one of a specific set of unwanted strings (e.g. "I'm sorry", "As an AI"). For hamful strings, the attack

Table 1: Comparison of various attacks in the harmful string setting

| Method | Proxy model | Target model | Success rate |
|---|---|---|---|
| GCG Pure Transfer [36] | Vicuna 1.3 13B | Vicuna 1.3 7B | 0.000 |
| GCQ (ours) | Vicuna 1.3 13B | Vicuna 1.3 7B | 0.388 |
| GCG Pure Transfer [36] | Vicuna 1.3 7B | Vicuna 1.3 13B | 0.000 |
| GCG Pure Transfer [36] | Vicuna 1.3 7B | Mistral 7B Instruct v0.3 | 0.000 |
| GCG Pure Transfer [36] | Vicuna 1.3 7B | Gemma 2 2B | 0.000 |
| GCQ (ours) | Vicuna 1.3 7B | Vicuna 1.3 7B | 0.791 |
| AutoDAN GA [19] | N/A | Vicuna 1.3 7B | 0.002 |
| AutoDAN HGA [19] | N/A | Vicuna 1.3 7B | 0.000 |

is successful only if the generation exactly matches the desired target string. Since the loss used by AutoDAN is the same as in GCQ (probability of generating the target string), we leave the loss unchanged. In this experiment, using the default repository parameters AutoDAN scored 1/574 and 0/574 in GA and HGA mode respectively, despite using much longer adversarial suffixes (around 70 tokens) compared to GCQ (20 tokens). In terms of query usage, the default parameters of AutoDAN correspond to about 128 iterations of GCQ.

We also evaluate GCG in the pure transfer setting of [36]. In this setting, we optimize the prompt purely against the proxy model, then evaluate the final string using the target model. We show the results in Table 1. In general, the low numbers for other attacks highlight how difficult it is to elicit specific harmful strings from models with a low degree of access.

## 4.3 Harmful strings for GPT-3.5 Turbo

We report results attacking the OpenAI text-completion model `gpt-3.5-turbo-instruct-0914` using GCQ. For our parameters, we used sequence length 20, batch size 32, proxy batch size 8192, and buffer size 128. We used the harmful string dataset proposed in [36]. For each target string, we enforced a max API usage budget of $1. For our proxy model, we used Mistral 7B [17]. Note that Mistral 7B is a base language model which has not been aligned, making it unsuitable as a proxy for a pure transfer attack. Using the initialization described in Section 3.2.3, we found that 161 out of the 574 (or about 28%) of the target strings were solved immediately, due to the model's tendency to continue repetitions in its input. Our total attack cost for the 574 strings was $80.

We visualize the trade-off between cost and attack success rate in Figure 2a. We note that the attack success rate rises rapidly initially. We are able to achieve an attack success rate of 79.6% after spending at most 10 cents on each target. This number rises to 86.0% if we raise the budget to 20 cents per target.

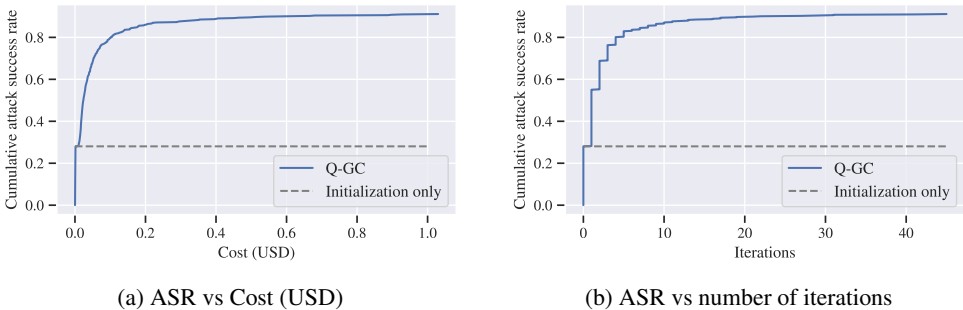

(a) ASR vs Cost (USD)        (b) ASR vs number of iterations

Figure 2: Attack success rate at generating harmful strings on GPT-3.5 Turbo, as a function of cost and iterations.

We also plot the trade-off between the number of iterations and the attack success rate in Figure 2b. The number of iterations corresponds to the amount of compute spent evaluating the proxy loss. This scales separately from cost because the cost of evaluating the loss using the API scales super-linearly

with the length of the target string, as we describe in Section 3.2.1, while the compute required to evaluate the proxy loss remains constant. Additionally, the short-circuiting of the loss described in Section 3.2.2 can cause the cost of the loss evaluations to fluctuate unpredictably.

**Analysis of target length.** We note that the attack success rates reported above are highly dependent on the length of the target string. We plot this interaction in Figure 3, which shows that our attack success rate drops dramatically as the length of the target string approaches and exceeds the length of the prompt. In fact, our success rate for target strings with 20 tokens or fewer is 97.9%. There are two possible reasons for this drop in success rate: *(1)* our initialization becomes much weaker if we cannot fit even one copy of the target in the prompt, and *(2)* we may not have enough degrees of freedom to encode the target string.

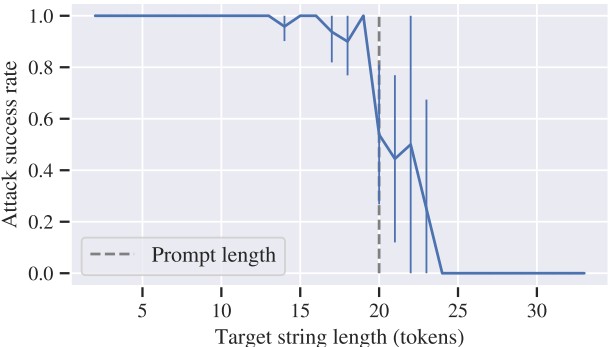

Figure 3: Tradeoff between attack success rate and target string length for a 20 token prompt. Attacks succeed almost always when shorter than the adversarial prompt, and infrequently when longer.

To demonstrate that this effect is indeed due to the length of the prompt, we ran the optimization a second time for the 39 previously failed prompts with length greater than 20 tokens using a 40 token prompt, which is long enough to fit any string from harmful strings. Since doubling the prompt length roughly doubles the cost per query, we upped the budget per target to $2. With these settings, we achieved 100% attack success rate with a mean cost of $0.41 per target. This suggests that longer target strings can be reliably elicited using proportionally longer prompts.

**Analysis of initialization.** To demonstrate the value of our initialization scheme, we perform an ablation where we instead use a random initialization. We reran our experiment for the first 20 strings from harmful strings, and in this setting, the attack was only successful only twice. This suggests that currently, a good initialization is crucial for our optimization to succeed in the low-cost regime.

### 4.4 Proxy-free harmful strings for open models

We evaluate the original white-box GCG attack, our optimized variant, and our optimized query-only variant from Section 3.3 on the task of eliciting harmful strings from Vicuna 7B. For each attack, we report cumulative success rate as a function of the number of attack queries to the target model's loss (in a setting where we only have access to logprobs and logit-bias, we can use the technique from Section 3.2.1 to compute the loss using black-box queries).

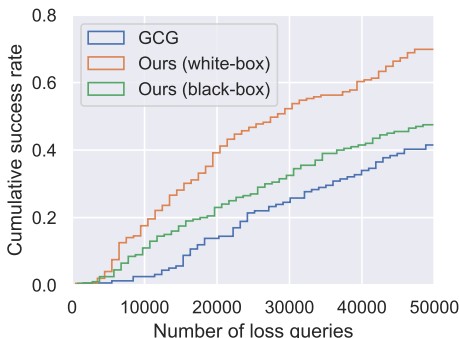

Figure 4: Our optimizations to the GCG attack require about $2\times$ fewer loss queries to reach the same attack success rate. When we remove the gradient information entirely to obtain a fully black-box attack, we still outperform the original GCG by about $30\%$.

Figure 4 displays the result of this experiment. Our optimized variant of GCG is approximately $2\times$ more query-efficient than the original attack, when gradients are available. When we sample token replacements completely at random, our fully black-box attack still outperforms the original GCG by

about 30%. Overall, this experiment suggests that black-box query-only attacks on language models can be practical for eliciting targeted strings.

### 4.5   Proxy-free attack on OpenAI `text-moderation-007`

One application of language models aims not to generate new content, but to classify existing content. One of the most widely deployed NLP classification domains is that of *content moderation*, which detects whether any given input is abusive, harmful, or otherwise undesirable. In this section, we evaluate the ability of our attacks to fool content moderation classifiers.

Specifically, we target the OpenAI content moderation model `text-moderation-007`, which OpenAI's "most robust moderation model to-date" [24]. The content moderation API allows one to submit a string and receive a list of flags and scores corresponding to various categories of harmful content. The scores are all in the range $[0, 1]$ and the flags are booleans which are True when the corresponding score is deemed too high and False otherwise. The threshold for the flags is not necessarily consistent across categories.

We demonstrate evasion of the OpenAI content moderation endpoint by appending an adversarially crafted suffix to harmful text. We consider the attack successful if the resulting string is not flagged for any violations. As a surrogate for this objective, we use the sum of the scores as our loss. This means we do not need to know what the category thresholds for each flag are, which is useful as they are not published online and may be subject to change.

As of February 2024, OpenAI does not charge for usage of the content moderation API, so we report cost in terms of API requests, which are rate-limited. For our evaluation, we use the harmful strings dataset [36]. Of the 574 strings in the dataset, 197 of them (or around 34%) are not flagged by the content moderation API when sent without a suffix. We set our batch size to 32 to match the max batch size of the API. We report results for suffixes of 5 and 20 tokens and for both nonuniversal and universal attacks.

**Universal attacks.**   In the universal attack, our goal is to produce a suffix that will prevent any string from being flagged when the suffix is appended. To achieve this, we randomly shuffle the harmful strings and select a training set of 20 strings. The remaining 554 strings serve as the validation set. We extend our loss to handle multiple strings by taking the average loss over the strings. The universal attack is more difficult than the nonuniversal attack for two reasons: *(1)* each evaluation of the loss is more expensive by a factor equal to the training set size (this is why we use a small training set) and *(2)* the universal attack must generalize to unseen strings.

For 20 token suffixes, our universal attack achieves 99.2% attack success rate on strings from the validation set, after 100 iterations (2,000 requests). We show learning curves across the duration of training to demonstrate the tradeoff between the number of queries and attack success rate in Figure 5b.

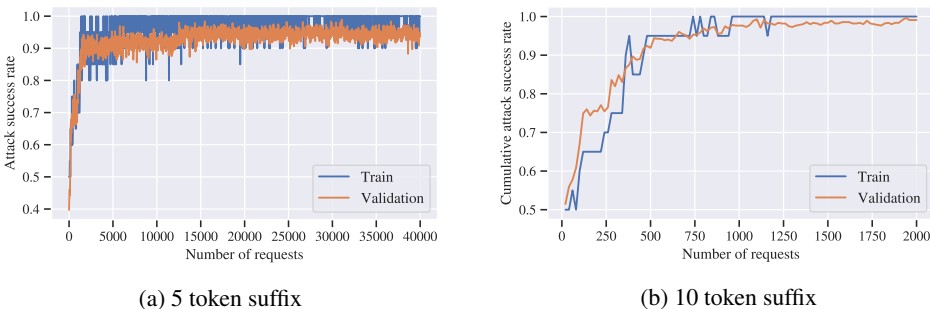

(a) 5 token suffix                                  (b) 10 token suffix

Figure 5: Universal content moderation attack success rate as a function of the number of requests for 5 and 20 token suffixes.

For 5 token suffixes, our universal attack achieves 94.8% attack success rate on strings from the validation set, after 2,000 iterations (40,000 requests). We show the corresponding learning curves in Figure 5a.

**Nonuniversal attacks.** In a nonuniversal attack, we are given a specific string which we wish not to be flagged. We then craft an adversarial suffix specifically for this string in order to fool the content moderator. We show the tradeoff between the maximum number of requests to the API and the attack success rate in Figure 6a. For 5 token suffixes, we find 83.8% of the strings receive no flags after 10 iterations of GCQ. For 20 token suffixes, that number rises to 91.4%.

### 4.6 Proxy-free attack on Llama Guard 7B

We attack the Llama Guard 7B content moderation model in the same setting as our nonuniversal OpenAI content moderation experiments. We show the results in Figure 6b. After 320 queries, the cumulative attack success rates for 5 and 20 tokens are 59% and 87% respectively, compared to 84% and 91% for OpenAI, and the gap between Llama Guard and OpenAI narrows with further iterations.

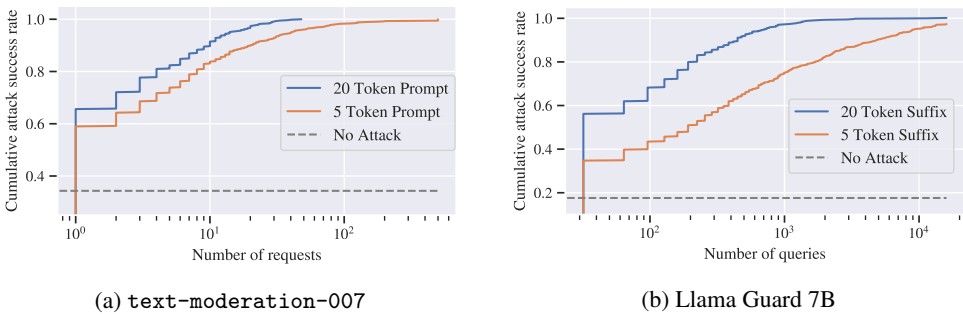

(a) `text-moderation-007`    (b) Llama Guard 7B

Figure 6: Nonuniversal content moderation attacks reach nearly 100% success rate with a moderate number of queries. Note that each OpenAI request corresponds to 32 queries.

## 5 Conclusion

In order to be able to deploy language models in potentially adversarial situations, they must be *robust* and correctly handle inputs that have been specifically crafted to induce failures. This paper has shown how to practically apply query-based adversarial attacks to language models in a way that is effective and efficient. The practicality of these attacks limits the types of defenses that can reasonably be expected to work. In particular, defenses that rely exclusively on breaking transferability will not be effective. Additionally, because our attack makes queries during the generation process, we are able to succeed at coercing models into emitting specific harmful strings—something that cannot be done with transfer-only attacks.

Although the attack we present may be used for harm, we ultimately hope that our results will inspire machine learning practitioners to treat language models with caution and prompt further research into robustness and safety for language models.

**Future work.** While we have succeeded at our goal of generating adversarial examples by querying a remote model, we have also shown that current NLP attacks are still relatively weak, compared to their vision counterparts. For any given harmful string, we have found that initializing with certain prompts can *significantly* increase attack success rates, while initializing with random prompts can make the attack substantially less effective. This is in contrast to the field of computer vision, where the initial adversarial perturbation barely impacts the success rate of the attack, and running the attack with different random seeds usually improves attack success rate by just a few percent.

As a result, we still believe there is significant potential for improving NLP adversarial example generation methods in both white and black-box settings.

## Acknowledgements

We are grateful to Andreas Terzis for comments on early drafts of this paper. JH is supported by the NSF Graduate Research Fellowship Program. This research was supported by the Center for AI

Safety Compute Cluster. Any opinions, findings, and conclusions or recommendations expressed in this material are those of the author(s) and do not necessarily reflect the views of the sponsors.

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

## A   Compute resources

For experiments in Section 4.1, we used between 2 and 8 A100 GPUs on a single node. The experiments took several days, although we did not have perfect utilization during that period. For our other experiments, we used a single A40 for several days.

## B   OpenAI logprob inference via logit bias and top-5 logprobs

As of March 2024, the OpenAI API does not allow the `logit_bias` parameter to affect the list of tokens returned in `top_logprobs`. This renders the following technique obsolete (at least for OpenAI models). We include it here for completeness.

As of February 2024, the OpenAI API supports returning the top-5 logprobs of each sampled token. By itself, this feature is not very useful for our purposes, since there is no guarantee that the tokens of our desired target string will be among the top-5. However, the API also supports specifying a bias vector to add to the logits of the model before the application of the log-softmax. This permits us to "boost" an arbitrary token into the top-5, where we can then read its logprob. Of course, the logprob we read will not be the true logprob of the token, because it will have been distorted by the bias we applied. We can apply the following correction to recover the true logprob

$$p_{\text{true}} = \frac{p_{\text{biased}}}{e^{\text{bias}}(1 - p_{\text{biased}}) + p_{\text{biased}}}.$$

The remaining challenge is to choose an appropriate bias. If the bias is too large, $p_{\text{biased}}$ is very close to 1, which causes a loss in accuracy due to limited numerical precision. On the other hand, choosing a bias that is too low may fail to bring our token of interest into the top-5.

In practice, we usually have access to a good estimate $\hat{p}_{\text{true}}$ of $p_{\text{true}}$ because we previously computed the score for the parent of the current string, which differs from it by only one token. Accordingly, we can set the bias to $-\log \hat{p}_{\text{true}}$ which avoids both previously mentioned problems if $\hat{p}_{\text{true}} \approx p_{\text{true}}$. If this approach fails, we fall back to binary search to find an appropriate value for bias. However empirically, our initial choice of bias succeeds over 99% of the time during the execution of Algorithm 1.

Unfortunately, the OpenAI API only allows us to specify one logit-bias for an entire generation. This makes it difficult to sample multiple tokens at once, because a logit bias that is suitable in one position might fail in another position. To work around this, we can take the first $i$ tokens of the target string and add them to the prompt in order to control the bias of the $(i+1)^{\text{th}}$ token of the target string. This comes with the downside of significantly increasing the cost to score a particular prompt: If the prompt and target have $p$ and $t$ tokens, respectively, then it would cost $pt$ prompt tokens and $t(t+1)/2$ completion tokens to score the pair $(p, t)$.

## C   Tokenization concerns

When evaluating the loss, it is tempting to pass the token sequences directly to the API. However, due to the way lists of token IDs are handled by the API, this can lead to results that are not reproducible with string prompts. For example, it is possible that the tokens found by the optimization are ["abc", "def"], but the OpenAI tokenizer will always tokenize the string "abcdef" as ["abcd", "ef"]. This makes it impossible to achieve the intended outcome when passing the prompt as a string. To avoid this, we re-tokenize the strings before passing them to the API, to ensure that the API receives a feasible tokenization of the prompt. We did not notice any impact on the success rate of Algorithm 1 caused by this re-tokenization.

Another concern is that the proxy model may not use the OpenAI tokenizer. Indeed, there are no large open models which use the OpenAI tokenizer at this time. To work around this, we also re-tokenize the prompts using the proxy model's tokenizer when evaluating the proxy loss.

## D   Defenses

There are many defenses that would effectively mitigate our attack as is, many of which are enumerated in [16]. Currently, our attack produces adversarial strings containing a significant number

of seemingly random tokens. Thus an input perplexity filter would be effective in detecting the attack. Incorporating techniques to bypass perplexity filters, such as those in [19] may give an effective adaptive attack for this defense. Additionally, our attack requires a method to estimate the log-probabilities of the model under attack for arbitrary output tokens. We believe effective attacks that work under the stricter black-box setting where log-probabilities cannot be computed is a promising direction for future work.

# E   OpenAI API Nondeterminism

Prior work has documented nondeterminism in GPT-3.5 Turbo and GPT-4 [25, 1]. We also observe nondeterminism in GPT-3.5 Turbo Instruct. To be more precise, we observe that the logprobs of individual tokens are not stable over time, even when the seed parameter is held fixed. As a consequence, generations from GPT-3.5 Turbo Instruct are not always reproducible even when the prompt and all sampling parameters are held fixed, and the temperature is set to 0. We do not know the exact cause of this nondeterminism. This poses at least two problems for our approach.

First, even if we are able to find a prompt that generates the target string under greedy sampling, we do not know how reliably it will do so in the future. To address this, we re-evaluate all the solutions once and report this re-evaluation number in Appendix E. Second, the scores that we obtain are actually samples from some random process. Ideally, at each iteration, we would like to choose the prompt with the lowest expected loss. To give some indication of the variance of the process, we plot a histogram of the loss of a particular prompt and target string pair sampled 1,000 times in Figure 7. We find that the sample standard deviation of the loss is 0.068. We estimate that our numerical estimation should be accurate to at least three decimal places, so the variation in the results is due to the API itself. In comparison, the difference between the best and worst elements of the buffer is typically at least 3, although the gap can narrow when very little progress is being made.

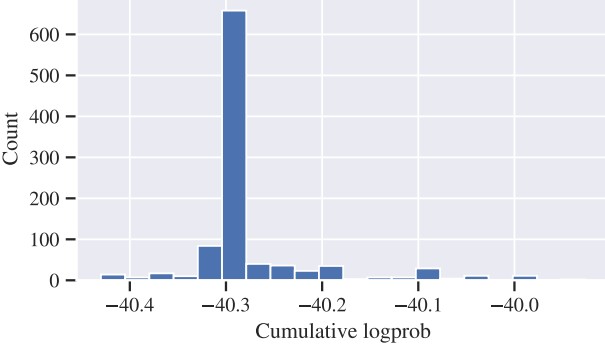

Figure 7: Histogram of cumulative logprob of a fixed 8 token target given fixed 20 token prompt, sampled 1,000 times.

We also found that the OpenAI content moderation API is nondeterministic. We randomly chose a 20 token input, and sampled its maximum category score 1000 times, observing a mean of 0.02 with standard deviation $4 \times 10^{-4}$. Because the noise we observed was relatively small in both cases, we decided not to implement any mitigation for nondeterministic losses during optimization, as we expect the single samples to be good estimators of the expected loss values.

**Nondeterminism evaluation.** To quantify the degree of nondeterminism in our results, we checked each solution an additional time. We found that 519 (about 90%) of the prompts successfully produced the target string a second time. This suggests that a randomly selected prompt will on average reproduce around 90% of the time when queried many times. We find this reproduction rate acceptable and leave the question of algorithmically improving the reproduction rate to future work.

