# OpenReview forum: "Query-Based Adversarial Prompt Generation"
_NeurIPS.cc/2024/Conference — NeurIPS 2024 poster_

### Official Review · Reviewer_67w9 · 2024-07-01

**Soundness:** 2
**Presentation:** 3
**Contribution:** 2
**Rating:** 5
**Confidence:** 4

**Summary:**

This paper proposes a query-based adversarial prompt generation method. It eliminates the prior attack's dependence on adversarial transferability and local surrogate models. The attack can evade the OpenAI and Llama Guard safety classifiers with a near 100% success rate.

**Strengths:**

## Originality
* The proposed improvement is interesting and novel.
* The related works are generally comprehensive.

## Quality
* Several practical considerations are included.
* The breadth of evaluations is good.

## Clarity
* The first three sections are generally well-written and easy to follow.

## Significance
* The proposed technique seems simple, but the overall contribution towards query-based attacks is important.

**Weaknesses:**

## Originality
I don't have major concerns here. The technical modifications of the existing attack GCG seem simple, but the good results can justify them. It is suggested to summarize and highlight the technical challenges.

## Quality

**Q: The evaluation setting is quite confusing.**

1. The paper was a bit vague regarding the attack's objective. Is it for a general jailbreaking attack (like GCG) or just eliciting the model to repeat a given harmful content? This also makes it hard to understand the attack success rate -- when do we count an attack as successful?

2. The evaluations do not match the major contribution. In Section 4, most of the evaluations are focused on the surrogate-based attack on various settings, including the actual GPT-3.5 model (a side question is why GPT-4 was not included since a black-box attack should work regardless of the underlying model). The actual contribution, the surrogate-free attack, was only evaluated on open models and then a different task of content moderation. The paper did not justify why the surrogate-based attack was not evaluated in the latter settings, and why the surrogate-free attack was not evaluated in the former settings. Ideally, the primary attack is supposed to be evaluated against all involved settings, and the surrogate-based attack is provided as a reference or ablation study.


## Clarity

* L33-45. It was a bit confusing when you first called out the surrogate-free property, then switched to a surrogate-based attack, and only finally, the further optimization that removes the surrogate model.
* L64-66. The high success rates in black-box attacks against classification tasks depend on two factors: the number of queries and the threshold perturbation. Most black-box attacks have 100% success rates at the first few queries but the inputs are either pure noise or heavily perturbed.
* Figure 1b. Please use different styles to distinguish curves of different groups of proxy or target. Right now, it is very hard to tell what the figure conveys and what we can learn when comparing curves with the same proxy size.

## Significance
N/A

**Questions:**

See weaknesses.

---

> ### Author Rebuttal · Authors · 2024-08-07
>
> We would like to thank the reviewer for carefully evaluating our work. We apologize for any confusion we may have caused, and will thoroughly review the presentation of our paper to avoid confusing future readers.
>
> ## Originality
>
> Following your suggestion, we will carefully revise the introduction to highlight the main contributions of our work. In particular, we will highlight the practical challenges that make black-box harmful string attacks difficult to run in practice, since overcoming these obstacles represents a significant part of our technical contribution.
>
> ## Attack objective
>
> The objective of the attack is to find a prompt that causes the model to output a specific string exactly. The attack is thus successful when the model generates exactly the desired string (including capitalization, punctuation, etc.). Thus the task is much more restrictive than jailbreaking. We elaborate on this task in the general rebuttal and we will add additional emphasis to the paper to clarify this point.
>
> [1] Zou, Andy, et al. "Universal and Transferable Adversarial Attacks on Aligned Language Models." (2023).
>
> ## Inclusion of GPT-4
>
> The reason GPT-4 was not included in our results is because it does not support logprobs. Although the attack may still be possible without logprobs, the expense would be beyond our budget.
>
> ## Evaluations and contribution
> The reason for this split is because it is best to use the surrogate when it is available (which is only under certain circumstances). Note that in the language modeling experiments, we are using a base pretrained model as the surrogate. Such surrogates are freely available and thus it makes sense to use them. However the downside is that they do not reveal anything about the alignment of the target black-box model. This is acceptable because we are only using the surrogate in a weak sense: to pre-filter the candidates before they are sent to the black-box model. We find that this weak pre-filter is still better than the position-based pre-filter used in the surrogate-free setting.
> However in some settings, such as in our content moderation examples, we may not have access to any kind of surrogate. Thus in this setting we apply the surrogate-free algorithm. Unfortunately we cannot afford to run the surrogate-free attack for closed models due to budget constraints, since we believe it will be much worse (i.e. much more expensive) than the attack using the weak surrogate. On the other hand, we cannot run the weak surrogate algorithm because the pretrained model surrogate does not make sense in the context of content moderation.
>
> You are correct that this arrangement appears awkward to readers. We will revise the discussion of the surrogate-free attack to make it more clear why the two attacks are applicable in different situations and we will aggregate the results common to both algorithms in Table 1.

---

> > ### Comment · Reviewer_67w9 · 2024-08-11
> >
> > Thanks for the response. Below is my understanding of the clarified contributions and evaluations, and my recommendation for adjusting the claims:
> > 1. The main contribution remains a query-based attack and an effort to reduce the dependency on transferability.
> > 2. The main technique is a better pre-filter based on an unaligned model; most evaluations are in this setting. However, this setting may not entirely remove the dependency on transferability, in the sense that it cannot be called surrogate-free or proxy-free attacks.
> > 3. In some settings, the model-based filter can be eliminated, but the resulting surrogate-free attack is generally too expensive to run. Therefore, the claims of surrogate-free attacks should be tuned down.
> >
> > After re-evaluating the paper based on the above breakdown, I'm borderline okay with the novelty justified by the performance, and strongly recommend the authors better articulate the technical challenges of 2.

---

### Official Review · Reviewer_CNWy · 2024-07-11

**Soundness:** 1
**Presentation:** 3
**Contribution:** 2
**Rating:** 5
**Confidence:** 4

**Summary:**

This paper modifies GCG, an attack on LLMs to elicit harmful responses, to create a query-based black-box attack with two primary goals: 1) Enable targeted attacks that are not possible with simple transfer-based attacks and 2) Enable attacks to still occur when no feasible surrogate model exists. To modify GCG, the key change is that the authors maintain a min-max heap of possible adversarial suffix candidates, searching for word replacements on the best currently known suffix candidate. The authors also experiment with prompt initialization, finding that including the target as many times as fits to be useful. Finally, as a query-optimizer in a proxy-free setting with random tokens instead of gradients, the authors experiment with trying a word replacement once in each position, and then taking the best position and trying more word replacements there. The authors find they are able to generate targeted attacks, which transfer attacks cannot, and can attack real-world models with proxy-free attacks.

**Strengths:**

Strengths include 1) enabling targeted attacks to elicit specific phrases, 2) enabling proxy-free attacks on real-world models without a surrogate, and 3) including optimizations to manage the query budget in the proxy free setting. Thus, the paper makes strides towards more practical attacks with adversaries with less power. The authors have also found their initialization strategy to help improve attack quality.

**Weaknesses:**

The primary weaknesses of this paper include a lack of comparisons against other black-box attacks (e.g., Andriushchenko et al., "Jailbreaking Leading Safety-Aligned LLMs with Simple Adaptive Attacks", and Lapid et al., "Open Sesame! Universal Black Box Jailbreaking of Large Language Models") on the proxy-free attacks. This also limits the technical novelty, as the attack as mostly a series of small changes to GCG. The other weakness is that this paper does not evaluate a wide variety of different types of LLMs - one of the themes is that the proposed attack is useful in settings where there is no good surrogate, so it would be nice to see how the attack does against transfer attacks across different families of LLMs.

**Questions:**

How does the attack perform against Andriushchenko et al. and Lapid et al.?
How does the attack do against transfer attacks across different families of LLMs?

**Limitations:**

Adequately addressed.

---

> ### Author Rebuttal · Authors · 2024-08-07
>
> We would like to thank the reviewer for carefully assessing our work. We hope the following will address all outstanding concerns and we are happy to further discuss during the discussion period.
>
> ## Comparisons against other black-box attacks
> Unfortunately, there is a dearth of works to compare to when it comes to eliciting exact harmful strings. The attack was originally proposed in GCG, where it was demonstrated against open models, but since then it has received little attention. This may be because there was no known method to cheaply compute the loss required for the harmful string attack under the constraints of popular APIs.
>
> Regarding the two suggested attacks:
>
> Andriushchenko et al. only focuses on controlling the first token generated after the prompt. This is because the first log-probability is directly given by OpenAI’s API. This is sufficient to elicit certain harmful behaviors, but not harmful strings of arbitrary length.
>
> Lapid et al. focus on jailbreaking, using a loss based on the embedding similarity to some target phrase, which is typically a short affirmation such as “Sure, here is”. This approach does not adapt well to harmful string targets because of the many local minima in the embedding similarity landscape.
>
> To demonstrate this, we repurposed GCQ to try to maximize the embedding similarity between an input string and a target string from harmful strings using OpenAI’s text-embedding-ada-002 model. In each of the 10 cases, we tried, the search terminated in a local minima, finding a string with a similar embedding but not the exact target string itself. (Note that this is much easier than solving the same problem indirectly through the [prompt → generation] mapping of an aligned LLM as needed for a harmful string attack.) On the other hand, the use of the embedding distance makes perfect sense for jailbreaking, where the target semantics are all that matter.
>
> The difficulty here is that neither of these attacks are designed to operate in our setting. We elaborate on this setting and why it is important in our general rebuttal.
>
>
> ## Technical novelty
>
> Although the actual optimization algorithm does require only minor changes to GCG. We would like to suggest that the computation of the loss itself is a significant technical contribution in its own right. To our knowledge, our attack is the only one which is able to elicit exact harmful strings (spanning many tokens) from a proprietary model, which is much more difficult than jailbreaking or eliciting harmful behaviors. The key to this ability is a procedure to quickly and inexpensively score the conditional probability of an entire sequence of generated tokens. Indeed one of the primary changes we make to the optimization algorithm (the introduction of the prompt buffer) is purely to help reduce the cost of the loss calculation.
>
> ## Transfer between model families
>
> We evaluate GCG in the pure-transfer setting within the same model family (in Table 1) and find that it performs very poorly (0 ASR) in the harmful string setting. This is in contrast with harmful behaviors, where transfer works quite well. Given the additional difficulty when transferring across model families, we would not expect it to work well in that setting. For example, we find that none of the strings computed for Vicuna 7B transfer to either Mistral 7B Instruct v0.3 or Gemma 2 2B Instruct.

---

> > ### Comment · Reviewer_CNWy · 2024-08-13
> > **Response to rebuttal**
> >
> > The author responses to my questions seem fair. I have updated my score accordingly.

---

### Official Review · Reviewer_ReRa · 2024-07-11

**Soundness:** 4
**Presentation:** 3
**Contribution:** 3
**Rating:** 7
**Confidence:** 4

**Summary:**

The paper presents a novel query-based attack method designed to generate adversarial examples that induce harmful outputs in aligned language models. Building on the GCG attack by Zou et al. (2023), this method employs a query-based strategy that eliminates the need for transferability, resulting in a significantly higher success rate. The effectiveness of the attack is demonstrated on GPT-3.5, even in black-box scenarios. Additionally, the paper details universal and prompt-specific attack techniques that effectively circumvent OpenAI’s safety classifier, achieving near-perfect evasion rates.

**Strengths:**

+ Despite the work from Zou et al. (2023) disclosing the vulnerability of production LLMs to adversarial examples capable of producing harmful strings and behavior, this paper overcomes some limitations related to attack transferability. The authors introduce a novel approach that leverages the API of production LLMs, like GPT-3.5, significantly improving the success rate of attacks and demonstrating the fragile behavior of LLMs in the presence of a smart adversary.

+ The paper is well-written, clearly describing each step of the attack, including strategies to bypass OpenAI’s API restrictions. For instance, the method for calculating the log probabilities is particularly smart and well-elaborated.

+ Overall, the experimental evaluation is comprehensive and provides valuable insights. For example, the results in Table 1 shows the limitations of previous attacks like CGC and AutoDan, whereas the attacks proposed by the authors are very effective both in white-box and transfer attack settings.

+ The authors also analyzed the effect of universal attacks, revealing interesting results in terms of the trade-off between the budget/cost and the effectiveness of the attack.

**Weaknesses:**

+ Although it is a relevant attack targeting a commercial system like ChatGPT 3.5, the scope of the attacks is somewhat limited as it requires API access to provide the logprob and only works with GCG. Nevertheless, I think that the paper clearly shows the risks of providing access to the logprobs in the API.

+ The paper only focuses on the attacker’s perspective. Defenses or possible mitigations are not properly discussed in the paper.

**Questions:**

+ Can the authors explain why the attack success rate of AutoDan are so low in the experiments reported in Table 1?

+ Can the authors provide any intuition about how to defend against an attack like this?

**Limitations:**

The limitations have been addressed appropriately in different sections of the paper, including, Section 3.2.1, and Appendix B, for example.

---

> ### Author Rebuttal · Authors · 2024-08-07
>
> We would like to thank the reviewer for their helpful comments! We hope that the following will address any residual concerns and we are happy to further discuss during the discussion period.
>
> ## Performance of AutoDAN
>
> The reason that AutoDAN performs so poorly is because it is being evaluated in an “unfair” way (out of necessity). The original AutoDAN code is designed to perform jailbreaking which is coded in roughly the following way: {the attack is successful if the model does not generate any of the following strings: (“No”, “Sorry”, “As an AI”, …]}. We replaced this objective with the objectives we used in GCQ: {The attack is successful if the model generates exactly “<example harmful string>”}. Because the harmful string objective is much more restrictive than the jailbreaking objective, the optimization of AutoDAN did not perform well. Of course this is reasonable since AutoDAN was not designed to work in this setting.
>
> We will adjust the presentation of this result to stress that the evaluation of AutoDAN is in a new setting where it cannot be expected to succeed. We hope that including this result will underscore that, in general, jailbreaking methods may not generalize to more difficult tasks, such as harmful strings.
>
> ## Defenses
>
> In terms of defenses, there are a variety of techniques that target various aspects of the text generation pipeline:
>
> The prompt can be filtered for e.g. high perplexity inputs (proposed in [1]). The attack we describe does often produce seemingly random sequences of tokens, so we would need to add additional functionality (for example, perplexity regularization) to evade this defense.
> The model generations can be filtered for harmful content by a second model. However as we show, it is possible to find strings that bypass popular content moderation models. Thus one could imagine a two-stage attack, where we optimize to generate a harmful string which also passes through the content filter.
> A different kind of defense would be to rate-limit queries that are very similar to previous queries (for example, if the API receives thousands of strings which all differ from each other by a small number of tokens). However, this has the potential to harm honest users, so it cannot be too harsh.
> Of course there is also the option of restricting access to logprobs, logit_bias or both as you note.
> Many more possible defenses are listed in [1].
>
> Of course, the best defense will always be a multilayered approach. Each layer of defense makes the attack much more difficult, as it is difficult to evade many defenses at the same time. We will add a section to the paper that discusses possible defenses in the spirit of what we've written here.
>
> [1] Jain, Neel, et al. "Baseline defenses for adversarial attacks against aligned language models." arXiv preprint arXiv:2309.00614 (2023).

---

> > ### Comment · Reviewer_ReRa · 2024-08-14
> >
> > Thank you very much for your comments. I think that the paper makes a nice contribution and I'm keeping my positive score.

---

### Official Review · Reviewer_A7Jt · 2024-07-12

**Soundness:** 3
**Presentation:** 3
**Contribution:** 2
**Rating:** 6
**Confidence:** 4

**Summary:**

This paper delves into the topic of adversarial examples and prompt injection attacks on aligned language models. A new strategy (GCQ) for black-box adversarial attacks is proposed which does not need access to a surrogate model, but only uses black-box access to the target model. This strategy is an extension of GCG, the current state-of-the-art attack. The proposed method works against both generative models and safety classifiers, as the experiments show.

**Strengths:**

- The experimental protocol covers a range of setups, including white-box, black-box, and universal attack.
- The main novelty of the paper is the no-proxy version of GCQ.
- The proposed attack GCQ shows good practical performance, in both white- and black-box setups, when compared to the existing state-of-the-art attack GCG and AutoDAN.
- Implementation included.

**Weaknesses:**

Significance:
- It would be great to include more attack baselines, e.g. AutoDAN attack from [Zhu et al., 2023].
- One of the most relevant qualities of the method is the independence of surrogate models. It seems relevant to compare against other attacks from the same category, e.g., PAL (cited as [28] by the paper, but not used as baseline).
- It also makes sense to include the GCQ without proxy access in Tab. 1.
- The topic of performance of the attack against existing defenses is not covered.

Novelty:
- For the GCQ version that still relies on a surrogate, the changes w.r.t. GCG seem minor.

Minor:
- Additional proofreading seems necessary.
- Sec. 4.2 and 4.6 are missing from the outline at the beginning of Sec. 4.
- L217 "hamful" -> "harmful"

## References

- [Zhu et al., 2023] Sicheng Zhu, Ruiyi Zhang, Bang An, Gang Wu, Joe Barrow, Furong Huang, Tong Sun. AutoDAN: Automatic and Interpretable Adversarial Attacks on Large Language Models. https://openreview.net/forum?id=ZuZujQ9LJV

**Questions:**

How does the GCQ attack perform against existing defenses?

**Limitations:**

Ok

---

> ### Author Rebuttal · Authors · 2024-08-07
>
> We would like to thank the reviewer for carefully reading our work and providing useful and constructive feedback. We hope that the following will address the points raised and are happy to further discuss during the discussion period.
>
> ## Comparison to other attacks
>
> Unfortunately, there is a lack of existing works to compare to when it comes to the harmful string attack. This attack was originally proposed in GCG, where it was demonstrated against open models, but since then, it has received little attention. This may be because there was no known method to cheaply compute the loss required for the harmful string attack under the constraints of popular APIs.
>
> Regarding the two suggested attacks:
>
> 1. The AutoDAN attack from [Zhu et al., 2023] is focused on jailbreaking under the constraint of an input perplexity filter. They do not evaluate in the setting of harmful strings and their optimization procedure is carefully tailored to the jailbreaking objective. Thus, it is not clear how to modify their method to elicit a specific string. Additionally they do not provide code, which makes the evaluation of their method difficult.
> 2. Similarly, PAL [28] focuses on Jailbreaking in the black-box setting. They use a similar collection of tricks to our work (developed independently and concurrently) to make the loss calculation feasible for commercial APIs, although the loss itself is different due to the different attack objective. Since PAL is also based on GCG, running PAL with the objective of GCQ would be very similar to running GCQ itself. Thus it is difficult to find a comparison that makes sense here.
>
> This underscores the lack of existing work on the harmful string attack. We believe this attack is meaningful and important, as we outline in the general rebuttal, and our hope is to make some initial progress in this direction beyond GCG.
>
> Regarding GQC with no proxy, we will add the results to Table 1 as requested. The numbers for Vicuna 7B were already computed for Section 4.4 so it is a simple matter to copy them over.
>
> ## Novelty
>
> Although the actual optimization algorithm does require only minor changes to GCG. We would like to suggest that the computation of the loss itself is a significant technical contribution in its own right. To our knowledge, our attack is the only one which is able to elicit exact harmful strings (spanning many tokens) from a proprietary model, which is much more difficult than jailbreaking or eliciting harmful behaviors. The key to this ability is a procedure to quickly and inexpensively score the conditional probability of an entire sequence of generated tokens. Indeed one of the primary changes we make to the optimization algorithm (the introduction of the prompt buffer) is purely to help reduce the cost of the loss calculation.
>
> ## Defenses
> We will add a section discussing defenses to the paper. In general, we expect defenses (e.g. input filtering or output filtering) to be effective against our attack. Since the attack is already quite difficult, raising the difficulty with defenses may make the attack infeasible unless new techniques are developed to further strengthen the attack.
>
> ## Presentation
> We will carefully proofread the paper and revise the outline to include all major sections of the paper.

---

### Author Rebuttal · Authors · 2024-08-07

We would like to thank all of our reviewers for their insightful comments. In the general rebuttal we would like to elaborate a bit on the harmful string attack.

## Harmful string attacks

In our language modeling results, we focus on the harmful string attack. This attack objective is to get the model to output a specific string exactly (including punctuation and capitalization for example). This attack is separate from jailbreaking or elicitation of harmful behaviors in that the desired outputs are highly specific. Empirically, this makes the attack much more difficult.

At first glance the harmful string attack may appear contrived, but it is significant for several reasons. First, it's a very straightforward test of model alignment. If a model creator's desire is “don't say X”, making the model say exactly X is a clear violation of those preferences. Second, it is easy to evaluate compared to jailbreaking which often must be manually evaluated by humans. Finally, exact strings can do harm in unique ways. For example a model supporting tools may be fooled into invoking the tool in a specific way which is harmful to the user.

---

### Decision · Program_Chairs · 2024-09-25

**Decision:**

Accept (poster)

**Comment:**

This paper presents a jailbreaking attack that's effective against safety classifiers and LLMs that produce log probs. Rather than trying to generate a harmful output in general, it attempts the narrower goal of generating a specific harmful string. All reviewers are positive about the contributions of this paper. Its greatest limitation is the reliance on log probabilities, which renders it ineffective against some commercial LLMs (notably GPT 4). It's also difficult to compare to some prior work, such as AutoDAN, because the harmful string objective is different and more specific. Finally, the authors do not demonstrate effective defenses; however, demonstrating effective defenses is difficult with LLMs, and many adversarial ML defenses have been shown to be ineffective when faced with an adaptive attacker. Overall, this paper is a positive contribution that better illuminates the attack landscape against LLM safeguards.